

# Size-dependent functional response of *Xenopus laevis* feeding on mosquito larvae

Corey J. Thorp[1], Mhairi E. Alexander[1,2], James R. Vonesh[1,3,4] and John Measey[1]

[1] Centre for Invasion Biology, Department of Botany & Zoology, Stellenbosch University, Stellenbosch, South Africa
[2] Institute for Biomedical and Environmental Health Research (IBEHR), School of Health and Life Sciences, University of the West of Scotland, Paisley, UK
[3] Department of Biology, Virginia Commonwealth University, Richmond, VA, USA
[4] Center for Environmental Studies, Virginia Commonwealth University, Richmond, VA, USA

Corresponding author
John Measey, john@measey.com

## ABSTRACT

Predators can play an important role in regulating prey abundance and diversity, determining food web structure and function, and contributing to important ecosystem services, including the regulation of agricultural pests and disease vectors. Thus, the ability to predict predator impact on prey is an important goal in ecology. Often, predators of the same species are assumed to be functionally equivalent, despite considerable individual variation in predator traits known to be important for shaping predator–prey interactions, like body size. This assumption may greatly oversimplify our understanding of within-species functional diversity and undermine our ability to predict predator effects on prey. Here, we examine the degree to which predator–prey interactions are functionally homogenous across a natural range of predator body sizes. Specifically, we quantify the size-dependence of the functional response of African clawed frogs (*Xenopus laevis*) preying on mosquito larvae (*Culex pipiens*). Three size classes of predators, small (15–30 mm snout-vent length), medium (50–60 mm) and large (105–120 mm), were presented with five densities of prey to determine functional response type and to estimate search efficiency and handling time parameters generated from the models. The results of mesocosm experiments showed that type of functional response of *X. laevis* changed with size: small predators exhibited a Type II response, while medium and large predators exhibited Type III responses. Functional response data showed an inversely proportional relationship between predator attack rate and predator size. Small and medium predators had highest and lowest handling time, respectively. The change in functional response with the size of predator suggests that predators with overlapping cohorts may have a dynamic impact on prey populations. Therefore, predicting the functional response of a single size-matched predator in an experiment may misrepresent the predator's potential impact on a prey population.

## INTRODUCTION

Predator–prey interactions are important in regulating prey populations and determining the structure of aquatic communities (*Brooks & Dodson, 1965*; *Carpenter, Kitchell &*

*Hodgson, 1985*). Predators directly impact prey populations by causing a decline in survival and recruitment, whereas prey quantity and quality directly affect feeding rate, growth, density, reproductive success and population dynamics of predators (*Miller et al., 1988*; *Luecke et al., 1990*; *Beauchamp, Whal & Johnson, 2007*). Consequently, these interactions can affect the distribution, habitat choice, behaviour and foraging strategies of both predators and prey (*Eggers, 1978*; *Sih, 1982*; *Walls, Kortelainen & Sarvala, 1990*). Classical predator–prey models typically assume that individual predators within a population are functionally equivalent (*Lotka, 1956*; *Volterra, 1928*; *Rosenzweig & MacArthur, 1963*). However, most species undergo considerable change in size during their ontogeny. Changing scaling relationships between predators and prey are known to produce nonlinear interactions, with intermediate size predators imposing the strongest *per capita* top-down interactions (*Vucic-Pestic et al., 2010*). Size differences of prey may have significant consequences for predator–prey interactions (*Jansson et al., 2007*; *Rudolf, 2008*; *McCoy et al., 2011*). The few studies that have quantified how predator size influences shapes of functional responses on the same prey have highlighted size-dependence of predator handling time and attack rate (e.g., *Eveleigh & Chant, 1981*; *Vucic-Pestic et al., 2010*; *Milonas, Kontodimas & Martinou, 2011*; *Anderson et al., 2016*), and even the general form of the functional response (*Anderson et al., 2016*). These studies show that assuming that predators of the same species are functionally equivalent may greatly oversimplify our understanding of within-species functional diversity and undermine our ability to predict predator effects on prey.

The functional response is the key relationship linking predator and prey dynamics and describes a predator's uptake of prey as a function of the prey density. *Holling (1965)* described the three most common models of predator functional response. A Type I response is characterized as having a constant attack rate $a$ with no handling time $h$ (*Holling, 1959*; *Hassell, 1978*). A Type II response incorporates handling time and, as a result, the rate of prey consumption by a predator declines at higher prey densities due to handling constraints. Handling time is the period predators are occupied with processing (e.g., ingesting, digesting) captured prey and are not able to engage new prey items. This constraint can produce nonlinearity to the relationship between prey availability and prey eaten. Predators that exhibit a Type II response typically de-stabilise prey populations due to the positive feedback on prey population growth caused by decreased predator consumption rates as a prey population increases, as predators are unable to regulate prey populations at densities beyond predator satiation (*Rosenzweig & MacArthur, 1963*; *Oaten & Murdoch, 1975*). A Type III response is defined by an accelerating increase in prey capture with increasing prey density for a range of low prey densities. The proportion of prey consumed initially increases with increasing prey availability then declines as in a Type II response (*Holling, 1959*; *Hassell, 1978*). This can create a refuge for prey at low densities, facilitating the persistence of prey populations, and a physical refuge in limited supply can create a Type III response. Therefore, the type of functional response a predator exhibits can result in quite different outcomes for prey. By describing the response, the potential impact at a population level may be elucidated (e.g., *Rosenzweig & MacArthur, 1963*).

Several factors may influence the type of functional response exhibited for a specific predator–prey interaction. This includes environmental conditions (e.g., *Laverty et al., 2015*; *Englund et al., 2011*) as well as body size of participants (e.g., *Brose et al., 2006*; *McCoy et al., 2011*; *Tucker & Rogers, 2014*; *Anderson et al., 2016*). Size variation is a common feature in animal populations and influences predator–prey interactions, competition and individual life histories (*Ebenman, 1988*; *Wilbur, 1988*; *Samhouri, Steele & Forrester, 2009*; *Asquith & Vonesh, 2012*). However, preferred prey typically change with ontogeny for many predators such that experiments are not able to present common prey across a range of predator sizes. For example, *Milonas, Kontodimas & Martinou (2011)* investigated the functional response of different instars of larval ladybirds (*Nephus includens*) using increasing prey sizes; all exhibited the same functional response type (Type II), but showed small differences in handling time and attack rate. For iteroparous amphibians with indeterminate growth and overlapping cohorts, individual body size is especially important *(Márquez, Esteban & Castanet, 1997*; *Werner, 1994)*. Smaller predators in these populations may be limited by the range of prey size they can consume (e.g., handling time may be greater for smaller predators; *Anderson et al., 2016*) and are often more efficient at assimilating consumed prey due to their high metabolic rates (*Werner, 1994*; *Asquith & Vonesh, 2012*). In contrast, their larger conspecifics are generally less efficient in converting prey biomass into predator biomass but may have a much broader range of prey sizes that they can consume (*Schoener, 1969*; *Asquith & Vonesh, 2012*; *Cohen et al., 1993*). In these populations, smaller predators may then have to deal with competition from larger predators that may result in a recruitment bottleneck that could potentially extend the period of time smaller predators remain at a vulnerable size (*Schröder et al., 2009*; *Asquith & Vonesh, 2012*). Therefore, understanding the relationship between consumer size and their feeding rates can provide insights into intra-cohort interactions and population dynamics of structured predator populations.

To investigate the role of predator size on functional response, we conducted a comparative functional response study between African clawed frogs, *Xenopus laevis*, of different sizes on a single prey type mosquito larvae, *Culex pipiens*, in order to answer the following questions: (1) Do differences exist in functional response type between different sized predators of the same species for a standardised prey size? (2) Are there differences in the functional response parameters (attack rate, handling time, and maximum feeding rate) of different sized predators?

## MATERIAL AND METHODS

### Study species

The focal predator species, the African clawed frog (*Xenopus laevis*, Daudin), has a wide distribution in southern Africa and inhabits permanent and temporary water bodies across its native range (*Measey, 2004*). In *X. laevis*, individuals within a population can vary as much as 8-fold in body size, with metamorphs as small as 15 mm snout vent length (SVL), to large adults exceeding 120 mm SVL (*De Villiers, De Kock & Measey, 2016*). *X. laevis* is a voracious predator with a broad diet that includes a wide variety of prey sizes and species,
ranging from vertebrates, such as adult frogs, to very small prey, such as zooplankton (*Vogt et al., 2017*; *Courant et al., 2017*).

*Culex pipiens* (Bedford), the northern house mosquito, is among the most widely distributed species of mosquito in the world (*Barr, 1967*). It is an important vector of St. Louis Encephalitis, West Nile Virus, Western Equine Encephalitis, Heartworm in dogs, and bird Malaria (*Turell, 2012*). *C. pipiens* breed in temporary surface-water habitats such as swamps, marshes, bogs, rice fields, and pastures, which can lack fish predators. Thus, *X. laevis*, which also utilize these temporary surface-water habitats and can readily disperse overland (*Measey, 2016*; *De Villiers & Measey, 2017*) to colonize newly formed aquatic habitats preferred by mosquitoes, may play a role in mitigating environmental health risk posed by this species.

### Specimen collection and maintenance

Adult *X. laevis* were captured in the field using funnel traps baited with chicken liver at the Jonkershoek fish hatchery (−33.9631°S; 18.9252°E), Western Cape Province, South Africa. All captured *Xenopus* were marked with a Passive Integrated Transponder (PIT) tag. *C. pipiens* were collected from naturally colonised populations using 50 l experimental tubs containing water and hay. Predators collected from Jonkershoek were transported to the Welgevallen Experimental Farm (−33.9426°S; 18.8664°E) where they were kept for a maximum of two weeks in ±500 l holding tanks. Predators were maintained on a diet of chicken livers *ad libitum*. All applicable institutional and/or national guidelines for the care and use of animals were followed, with ethics clearance for experiments granted by Stellenbosch University Research Ethics Committee: Animal Care & Use (SU-ACUD15-00011). Collection permits were granted by CapeNature (permit number AAA007-00159-0056).

### Experimental procedure

To quantify the functional response of *X. laevis* preying upon *C. pipiens* mosquitoes dependent on consumer body size, we conducted a 3 × 5 factorial experiment in which three size classes of predator were crossed with five prey densities in independent trials. Predators were classified into three size classes according to their snout vent length (SVL, mm; mean ± SD): small (21.0 ± 3.9), medium (54.6 ± 2.6) and large (113.3 ± 4.6). *C. pipiens* larvae used were size-sorted (7–9 mm thorax length) using mesh screening and were all likely fourth instar. Prey density treatments were 20, 50, 100, 200 and 500 larvae per ±500 l rectangular mesocosm, giving densities of 0.04, 0.1, 0.2, 0.4 and 1 mosquitoes per litre, respectively. Treatments with single predators were randomly assigned and replicated four times.

Experiments were conducted between 15 March and 13 May 2016 in individual ±500 l rectangular mesocosms placed outdoors in single block at the Welgevallen Experimental Farm, Stellenbosch. Mesocosms were rectangular plastic bins with a capacity of 1,000 l, half-filled with water to 50 cm depth (volume of approximately 500 l), and covered with mesh screening to prevent any disturbance. These frogs are active between evening and midnight of each day (*Ringeis et al., 2017*), while the mosquito larvae are suspended at the
surface at all times. Predators were placed into the mesocosms 24 h prior to experimental trials to acclimate. Hunger levels were standardised by not feeding *Xenopus* for 48 h prior to the experiment. Experiments were initiated at 18:00 with the addition of mosquito larvae and were completed once the predators were removed after 14 h at 08:00 the following day. Remaining prey were counted in order to determine the predator's functional response. During the experiment, we maintained a mesocosm with the highest density of prey, but without predators, to assess short-term background mortality or biases in recovery. We observed no mortality and recaptured all larvae from these controls. Thus, we assume background mortality from causes other than *Xenopus* predation in experimental trials was negligible.

## Statistical analysis

All functional responses were modelled in R v3.3.1 (*R Core Team, 2013*) using the ''friar'' package (*Pritchard et al., 2017*) via a two-step process.

First, we used logistic regressions to distinguish between Type III and I & II functional response types (*Trexler, McCulloch & Travis, 1988*; *Juliano, 2001*). To accomplish this, we modelled proportion of prey killed as a function of prey density. If the first-order term of the analysis was significantly negative, the functional response was considered a Type II. If the first-order term was significantly positive, followed by a significantly negative second-order term, the functional response was considered a Type III (*Juliano, 2001*).

Second, once we determined the general form, functional responses were fit using a flexible model that includes a scaling exponent $q$ to allow for a continuum of shapes between types I, II and III to be described (*Barrios-O'Neill et al., 2015*; *Real, 1977*):

$$N_e = N_0(1 - exp(bN_0^q(hN_e - T)))$$
(1)

where $N_e$ is the number of prey eaten, $N_0$ is the initial prey density, $b$ is the attack rate, $h$ is the handling time, $q$ is the scaling exponent and $T$ is the total time available. Where Type II responses occur, $q = 0$, and functional responses become increasingly Type III in form when $q > 0$. In order to compare functional responses of different size classes, 95% confidence intervals were fitted around functional response curves by non-parametrically bootstrapping the datasets ($n = 2,000$).

# RESULTS

## Functional response model

Logistic regression indicated that of the three size classes of *X. laevis,* small frogs clearly exhibited a Type II functional response, as revealed by the significantly negative first-order term (Table 1, Fig. 1). The scaling exponent, $q$, was therefore fixed at 0. Logistic regression indicated Type III responses for medium and large size classes (Table 1, Fig. 1). For these size classes, $q$ was unfixed for initial model fitting and then fixed at the generated maximum likelihood estimate. Bootstrapping was performed on the parameters $b$ and $h$ to provide an error estimate.

Table 2 provides estimates for the functional response parameters $b$ and $h$ for all size classes studied and $q$ in the case of medium and large *X. laevis*. The only differentiation

**Table 1** Parameter estimates from logistic regression analyses of proportion of prey (*C. pipiens*) consumed against initial prey density for small, medium and large size classes of *X. laevis* predators. Values for 1st order and 2nd order terms are presented with *p* values.

| Size class | Intercept (*p*-value) | 1st order (*p*-value) | 2nd order (*p*-value) | Functional response type |
|---|---|---|---|---|
| Small | 2.541 (<0.001) | −0.007 (<0.001) | – | II |
| Medium | −0.106 (<0.05) | 0.0045 (<0.01) | −0.000006 (<0.01) | III |
| Large | −1.494 (<0.001) | 0.0098 (<0.001) | −0.000015 (<0.001) | III |

occurring between functional response curves was at low prey densities (i.e., 0–100) where small frogs had higher predation rates compared to medium and large size classes (Fig. 1). This was supported by the higher attack rate for small size classes (Fig. 2A). Responses converged at higher densities between medium and large size classes as well as small and large size classes, with overlapping confidence intervals for the asymptotes (Fig. 1) and handling time coefficients (Fig. 2B) overlapping. Handling time coefficient was highest in the smallest predator size class, and lowest in the medium size class, with a significant difference (Table 2; Fig. 2B). Handling time for frogs in the largest size class was intermediate, and not significantly different between medium and small frogs (Fig. 2B).

## DISCUSSION

We found changes in the basic form of the functional response type between different sized predators of the same species for a standardised prey size. The smallest predator size class exhibited a Type II response compared to Type III responses as exhibited in medium and large adults. This finding has important implications for understanding how predator–prey dynamics change in systems where predators undergo large changes in body size relative to their prey through ontogeny. Moreover, we show predator attack rates and handling times change with predator size. Search efficiency was found to be inversely proportional to predator body size whereas handling time exhibited a U-shaped function and maximum feeding rate was observed in medium sizes of *X. laevis*. Predators of the same species are often assumed to be functionally equivalent, despite individual variation in predator traits known to be important for shaping predator–prey interactions, like body size (*McCoy et al., 2011*, but see *González-Suárez et al., 2011*). This assumption may greatly oversimplify our understanding of within species functional diversity and undermine our ability to predict predator effects on prey. Here we examine the degree to which predator–prey interactions are functionally homogenous across a natural range of predator body size.

Frequently, handling time initially decreases with increasing predator size, which can be attributed to an increased digestive capacity and gape size (*Mittelbach, 1981*; *Persson, 1987*). However, *Persson et al. (1998)* theorised that handling time will decrease until it reaches a minimum value, as found by *Mittelbach (1981)*, and at some point will begin to increase with predator size (e.g., *Persson, 1987*). This is consistent with our findings where medium sized predators were found to have the lowest handling time, potentially representing the minimum amount of handling time across all size classes. A possible explanation is that large predators will have difficulty in handling very small prey and small predators may

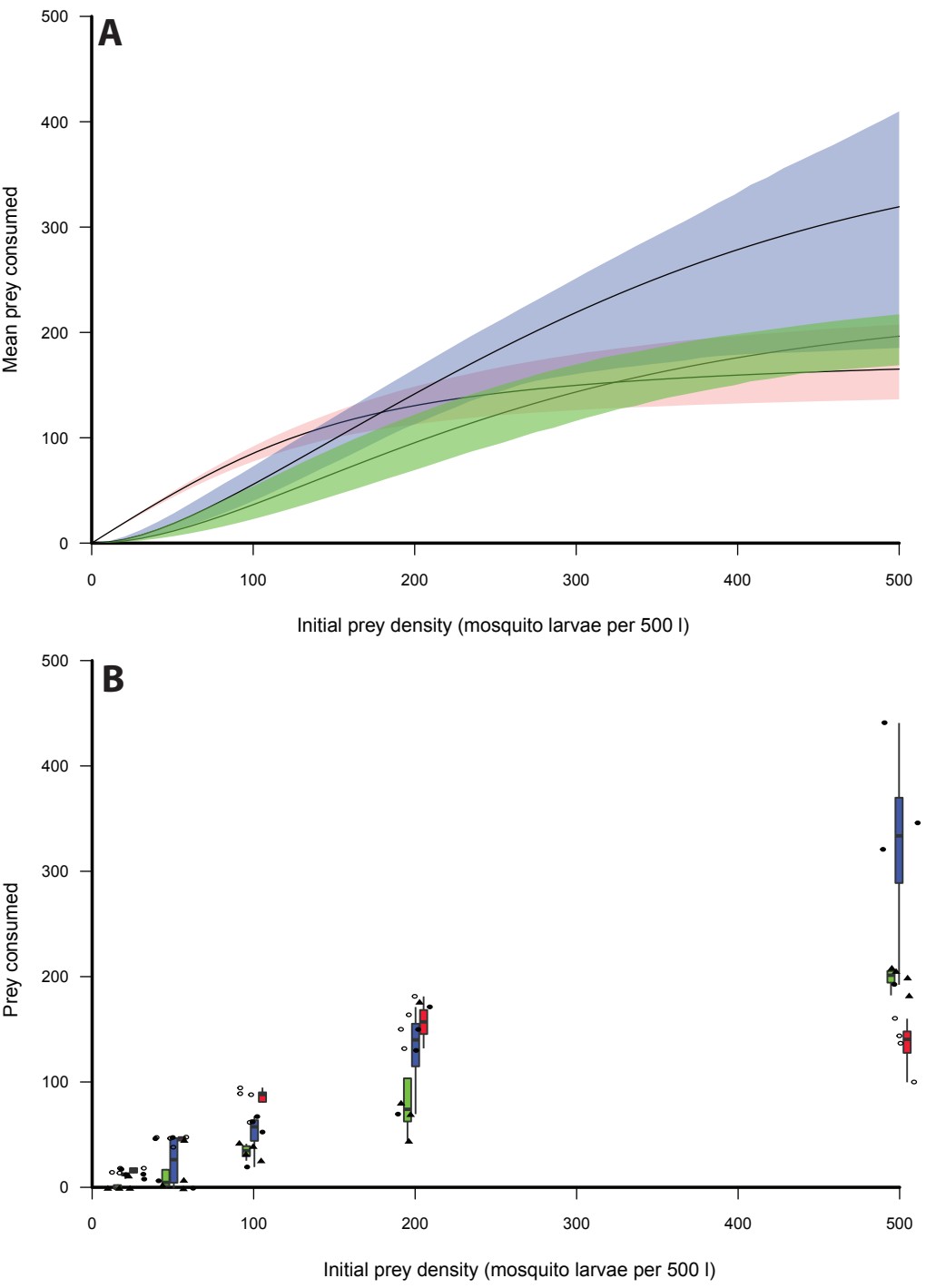

**Figure 1 Functional responses of *X. laevis* preying on mosquito larvae.** (A) Functional responses of individual small (red), medium (blue) and large (green) size classes of *X. leavis* in different initial densities of mosquito larvae (per 500 l). Solid lines represent model curve and shaded areas represent 95% confidence intervals calculated by non-parametric bootstrapping. (B) Box plots and data points for each trial with small (red, open circles), medium (blue, closed circles) and large (green, closed triangles) size classes of *X. leavis*.

**Table 2** **Results of the flexible functional response model to prey consumed by size classes of *X. laevis*.** Parameter estimates of search coefficient ($b$ in seconds), handling time ($h$ in seconds) and scaling coefficient ($q$) from fitting the flexible functional response model to prey (*C. pipiens*) consumed against initial density for small, medium and large size classes of *X. laevis*. Estimates presented with standard error.

| Parameter estimate | $b$ | $h$ | $q$ |
|---|---|---|---|
| Small | $3.526 \pm 0.202$ | $0.005 \pm 0.0001$ | Fixed at 0 |
| Medium | $0.212 \pm 0.064$ | $0.001 \pm 0.0003$ | $0.320 \pm 0.069$ |
| Large | $0.117 \pm 0.080$ | $0.004 \pm 0.0003$ | $0.738 \pm 0.109$ |

have an increased handling time due to their digestive capacity or the prey being large to ingest by inertial suction (*Persson, 1987*). Therefore, it might be expected that these larger predators will favour larger prey in order to increase their capture success rate. However, there are multiple examples in the literature that show *X. laevis* predators, independent of size, predominantly consume zoobenthos and zooplankton (*Courant et al., 2017*). This could be attributed to prey availability and density where the lower limit for prey size consumption depends on prey encounter rate and the cost of consumption (*Elton, 1927*; *Owen-Smith & Mills, 2008*). Very little movement is required to feed on both zooplankton and zoobenthos which would reduce energy cost and predation risk. Low densities of small prey offer very little reward to large predators which may explain why both medium and large sized predators did not consume high proportions of prey when prey density was low (*Griffiths, 1980*).

There are a number of examples that exist showing unimodal ('dome shaped') relationships between attack rate and predator size (*Aljetlawi, Sparrevik & Leonardsson, 2004*; *Tripet & Perrin, 1994*; *Werner, 1988*). In aquatic predators, the initial increase of attack rate with predator size is most likely due to an increase in predator search speed, which will positively affect prey encounter rates (*Keast & Webb, 1966*; *Schoener, 1969*). The eventual decline in attack rate with increasing predator size could be due to either prey being relatively too small to be detected or the inability of a predator to make fine-tuned movements, resulting in lower prey capture success rate (*Hyatt, 1979*). However, in our study, attack rate was not dome shaped with respect to prey size and instead negatively correlated with size class (Table 1). One explanation is that the dome shape may only be observed if the experiment had additional intermediate predator size classes. Therefore, attack rate may yet hold a dome shaped function of predator size, which may exist between the small and medium size classes measured in this study. Another explanation for the negative correlation could be that the prey are already at the optimal size for maximum attack rate in small sized predators. There is also a possibility that the relative fitness gain from small prey items is too small to make it worthwhile for larger foragers to be active.

*Milonas, Kontodimas & Martinou (2011)* found different feeding modes in a predatory ladybird (*Nephus includens*) in which smaller instars (2nd instar, 2 mm) were found to partially consume prey of different sizes, whereas larger instars (4th instar, 3.3 mm) consumed prey whole. The differences in feeding mode between the small and large predators led to differences in handling time when prey size was increased. Smaller predators were able to maintain a constant handling time, whereas larger predator's

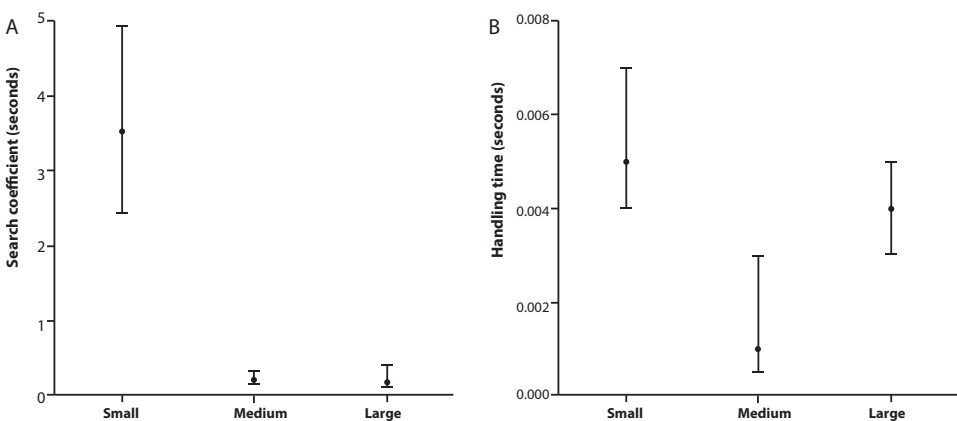

**Figure 2 Search coefficient and handling time from functional response models.** (A) Search coefficient (in seconds) and (B) handling time (in seconds) parameters derived from flexible functional response models for small, medium and large size classes of *X. laevis*. Points are original model values and error bars are bootstrapped 95% Confidence Intervals.

handling time increased with prey size. However, in our study all predators completely consumed prey; thus the mosquito larvae were not too large for the smallest frogs to consume. The lower capture success rate found in medium and large predators was most likely due to their limited ability to hold relatively small prey (CJ Thorpe, pers. obs., 2016), similar to observations made on fish (*Persson, 1987*). Observation data also showed a response from predators to movement from prey. Regardless of the predator's positioning in relation to the prey, detection was most likely when prey exhibited sudden movements. *X. laevis* do not principally use visual or olfactory cues in order to detect aquatic prey, and instead rely on their sensitive lateral line systems (see *Elepfandt, 1996*).

Despite the potentially profound implications for predator–prey dynamics, few studies directly test whether the basic form of the functional response changes with consumer size. Recently, *Anderson et al. (2016)* found that the form of the functional response changed with predator size (hatchling to larval ambystomatid salamanders), with smaller predators (adult ambystomatid salamanders) being more limited by handling times than large predators. In other words, smaller predators tended to exhibit a Type II functional response while larger predators exhibited a Type I functional response for the same prey. Type II functional responses as defined by Eq. (1), collapse to a Type I functional response when estimates of the handling time parameter overlap zero. None of the predators exhibited a Type III functional response. In this study, we find that both medium and large sized *X. laevis* showing a Type III response and small predators exhibiting a Type II, smaller predators may be able to exploit prey at low densities. There is a trend towards higher $q$ values (or scaling exponent) and a more stabilising response (*Alexander et al., 2012*).

Thus, the medium size class of *X. laevis* is most likely to destabilise predator–prey dynamics given fast handling times and a reduction in consumption at low densities as indicated by a lower $q$ than the large size class. Small frogs are likely to destabilise prey at low densities, but overall they have a much lower handling time, and therefore a higher feeding

rate. When prey density is low, there is an increase in predation from small predators, and when prey density is high, there would be an increase in predation from larger predators (*Rindone & Eggleston, 2011*). Densities of *X. laevis* are known to reach very high levels, especially in invasive populations (e.g., *Measey, 2001*; *Lobos & Measey, 2002*; *Faraone et al., 2008*), but also in natural assemblages (*De Villiers, De Kock & Measey, 2016*). The present study also has a conservation context as the smaller, but functionally similar, congener *X. gilli* is threatened by competition from *X. laevis* (see *Vogt et al., 2017*). Thus, having a population of predators of different sizes at the same time means that there is little relief for multiple prey species, and could lead to prey extirpation (*Hassell, 1978*). This could be advantageous, if the prey species is a potential disease vector, as in the case of *C. pipiens*. Prey may experience a similar scenario with fish in aquatic ecosystems due to many fish species consisting of populations with overlapping cohorts (*Werner, 1994*). However, in populations where differences in predator size are less pronounced, prey may experience only one type of predator response (*Milonas, Kontodimas & Martinou, 2011*).

## CONCLUSION

Studies often compare functional responses of native and invasive predators and important inferences are made about the potential impacts of these invaders (reviewed by *Dick et al., 2013*). However, little research focuses on the potential role predator size could play in determining these functional responses. Predators can change their foraging preference as they age and grow and selecting a single size class in functional response experiments to represent an entire population may not be the best representation of populations with overlapping cohorts and large size ranges. It is important to consider whether the same pattern would be seen on different prey species. How would functional response curves be affected if prey size was increased (e.g., see *McCoy et al., 2011*)? There may be a shift from a Type III to a Type II functional response in our medium and large sized predators as prey size increases. Similarly, it could be asked how prey traits (e.g., activity, shape, colour, etc.) affect functional response curves when size is kept constant. It is therefore important to answer these questions so that a predator population's functional response is correctly represented. This study has shown parameters such as attack rate, handling time and maximum feeding rate as well as functional response type are dependent on predator body size. Therefore, when conducting a functional response experiment it is vital to consider both predator and prey size, foraging strategy and prey species.

## ACKNOWLEDGEMENTS

We would like to thank members of the MeaseyLab for their help in preparation and harvesting of experiments: Erin Jooste, Ana Nunes, Giovanni Vimercati, Nitya Mohanty, Marike Louw, Mohlamatsane Mokhatla, Alex Rebelo. We thank Donald Kramer, Peter Abrams, and an anonymous reviewer for their constructive comments. We would like to thank staff at Welgevallen Experimental Farm for facilitating the experimental work.

### Funding

The National Research Foundation (NRF) of South Africa (NRF Grant No. 87759 to John Measey) provided financial support for the project and a bursary to Corey Thorp. James Vonesh was partially supported by the US Fulbright Fellowship Program and NSF DEB 1556686. All authors were supported by the DST-NRF Centre of Excellence for Invasion Biology. There was no additional external funding received for this study. The funders had no role in study design, data collection and analysis, decision to publish, or preparation of the manuscript.

### Competing Interests

John Measey is an Academic Editor for PeerJ.

### Author Contributions

- Corey J. Thorp conceived and designed the experiments, performed the experiments, analyzed the data, prepared figures and/or tables, authored or reviewed drafts of the paper, approved the final draft.
- Mhairi E. Alexander conceived and designed the experiments, analyzed the data, prepared figures and/or tables, authored or reviewed drafts of the paper, approved the final draft.
- James R. Vonesh conceived and designed the experiments, performed the experiments, contributed reagents/materials/analysis tools, authored or reviewed drafts of the paper, approved the final draft.
- John Measey conceived and designed the experiments, performed the experiments, contributed reagents/materials/analysis tools, authored or reviewed drafts of the paper, approved the final draft.

### Animal Ethics

The following information was supplied relating to ethical approvals (i.e., approving body and any reference numbers):

All applicable institutional and/or national guidelines for the care and use of animals were followed, with ethics clearance for experiments granted by Stellenbosch University Research Ethics Committee: Animal Care & Use (SU-ACUD15-00011).

### Field Study Permissions

The following information was supplied relating to field study approvals (i.e., approving body and any reference numbers):

Collection permits were granted by CapeNature (permit number AAA007-00159-0056).

### Data Availability

The raw data are available in a Supplemental File.

## Supplemental Information

Supplemental information for this article can be found online at http://dx.doi.org/10.7717/peerj.5813#supplemental-information.

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
