# Peer review of "Size-dependent functional response of Xenopus laevis feeding on mosquito larvae"

_PeerJ, doi:10.7717/peerj.5813_

## Round 0.1 · original submission · Major Revisions

This study examined the functional response of three size classes of Xenopus feeding on one size of mosquito larvae and estimated the components that produce the functional response from both curve-fitting and from a video analysis of feeding in a different experimental set up. The smallest size class showed a Type II response whereas the intermediate and larger size classes showed a Type III response. Parameters estimated from the curves indicated that the small size class had a substantially larger search coefficient than medium and large size classes and that handling time showed a trend for the medium size class to have shorter handling time than small and large, although the difference was statistically significant only between small and medium sizes. Estimates from the video analysis showed that attack rate and attack efficiency were higher in small than medium and large frogs and that handling time was shorter in medium than in small and large frogs. Although an objective was the comparison of attack rate and handling time derived from functional response curves and direct observations, no quantitative comparison of handling time was carried out and different variables (search coefficient and attack rate) were derived from the two approaches.

Reviewer 1 recommended minor revision and provided numerous comments to strengthen the context and interpretation of the study. He also noted the lack of attention to the third objective, comparing model-derived and observation-derived measures of functional response components. Reviewer 2 also recommended minor revision, noting the need for an appropriate introduction to the objective of comparing model- and observation-derived measurements and providing many helpful suggestions for clarification of the manuscript. Note that the detailed comments from Reviewer 2 were originally submitted as an annotated pdf. PeerJ staff moved them to the main review; the reviewer did not submit an annotated pdf of the manuscript itself.

I have concerns about the analysis of the observational data and the limited attention to the third objective, as well as numerous issues of clarity, consistency, organization and methodological completeness. Because the first two issues could change your conclusions, I consider the needed revisions major rather than minor. You can respond to my comments below as a third review, i.e. you are free to disagree if I am in error.

Editor's comments
1) Observational data analysis.
" First, several of the behavioral measures are not sufficiently well defined to allow another research to repeat the study. The entire paragraph starting on L201 should be clarified and organized more logically and sequentially. This may require starting with a general description of Xenopus feeding in your system with attention to the terms used. Include information about where the mosquito larvae are (mostly at the topic) and how much active and passive search the frogs do.
" Encounter rate is defined as the total number of predator-prey encounters divided by the experimental time period. This requires an operational definition of encounter. Furthermore, using the total experimental time period means that handling time as well as time in other activities (e.g. hiding, traveling to breathe at the surface) are assumed to be included in search which is not valid.
" For encounter rate, dividing density by volume of water implies that you are dividing by volume twice. Furthermore, I cannot see how density is equivalent to prey detection; it would make more sense for this to be related to encounter (with some measure assumptions about movement of predator and prey).
" For eating time, it is not clear how you defined ingestion, since the prey was presumably out of view.
" For attacking time, you need to specify what the initial lunge is, since the predator may be moving already at the time the attack is initiated. It needs to be clear how you defined an attack if it was unsuccessful.
" Your statement about the calculation of handling time (L210) by adding a ratio will be unclear to readers. It is not obvious that one can add a ratio to a time and get a time. Be clear that this is a procedure to add handling time for unsuccessful attacks.
" Your use of the term attack rate defined in L203 is unclear. Jeschke et al. called this predation rate which seems more appropriate because attack rate would be expected to include unsuccessful attacks as well. For the observational data, you record attack rate directly (sum of successful and unsuccessful attacks, divided by observation time minus handling time and other not searching time). Why would you then calculate it using all these separate and somewhat doubtful components?
" You should specify the camera speed and time conversion you used if your method involved counting frames. If you used some sort of time record, indicate its precision.
" Your supplementary data are quite incomplete, giving only a single value for each variable without any indication of the number of events of each type recorded.
" I emailed Reviewer 1 about these issues as he had not mentioned some of them, and I was wanted to make sure that I was not mistaken in my concerns. His reply follows: "I did mention that the authors had not defined when an encounter had occurred. They also did not seem to use this video data in a useful way--i.e. actually comparing handling time estimates from the FR relationship based on consumption measurements vs. direct observation of handling on video. There are several things strange about the description of parameter estimation using video. They have no need to break down that attack rate into the many components that they cannot properly measure. You are right about the need to subtract handling individuals in determining the number that could attack, but they may have sort of done this by not counting something as an encounter if it involved an individual that was handling so did not attack. If nothing else, this section on the video analysis needs a lot of clarification. I would recommend just using the video data to estimate two parameters--time spent in unsuccessful attacks and time spent in successful attacks. It would be interesting to know (and important for the relationship) to know whether the ratio of successful to unsuccessful changed over the course of the experiment or whether it differed between prey density treatments. Feel free to add the above to my review."
2) Objective 3
" Your goal of comparing model-derived and empirical components of the functional response is a worth-while one. However, the Introduction should develop the rationale for this objective, the Results should address it, and the Discussion should consider the implications of your findings. As also questioned by Reviewer 2, you are calculating a different component in the model (search coefficient which is area searched per unit time) and observational data (attack rate which you define as successful predation events per unit time). This needs to be rectified before attempting to compare the two approaches.
3) Units are missing in tables, figures, supplementary data and elsewhere (both methods and results). When you first define a variables you should indicate the units in which it will be measured and then provide those units throughout when you give the numbers.
4) Discussion organization and content. The Discussion should be organized to systematically follow the order of your objectives, including the sequence of components in objective 2. This will make it much easier for readers to follow the interpretation of your findings. It is important that you adopt a sufficiently self-critical stance to clearly distinguish between what you actually showed and how you interpret the patterns. You will see some of my concerns about this in my comments below on the first paragraph of the Discussion.
5) In your discussion of the functional response type, you put substantial emphasis on the difference in functional response type but less on the magnitude and range of the difference. The implications of ignoring size are not so drastic if the effect size is small and limited to only one range. Why might the small frogs have differed only at low densities, and is this important? Is there any significance to the similarity of medium and large frogs? Perhaps the discussion starting on L285 is relevant here. But note that you need to be careful in presenting as a clear finding what seemed to be an unquantified casual observation lacking statistical support.
6) The discussion of attack rate seems to conflate search rate with predation rate. They are not the same and may not respond to size the same way. Do Keast & Webb and Schoener really say that an increase of burst swimming speeds increase encounter rate? It seems quite unlikely because most foragers do not encounter prey while swimming at burst speeds (more appropriate for attacking a prey) but while moving at cruising speeds or sitting and waiting. Furthermore, this conclusion is based on cruising predators and I wonder if the frogs meet that assumption. If they are sitting on the bottom while searching, the model would not apply. Although you imply that the attack rate and search coefficient show similar patterns, the attack rate show a great similarity between size groups. Indeed, I am surprised at the highly significant difference as only 6 of the 15 small frogs were outside the range of the other two size classes according to your supplementary data and the effect size is very small. Indeed, given the high density situation in which they were tested, you might comment on why the larger frogs did not eat more prey than the smaller ones.
7) Your paragraph on handling time assumes that the U-shaped relationship is a valid interpretation. You need to establish the strength of this pattern before discussing it because the difference between medium and large frogs was not significant. Is there any way that your recording of the phases of prey ingestion could have been affected by frog size, that the handling time pattern is an artifact of the method? This is especially an issue with the extremely short handling times (even though we don't know if they are on the order of milliseconds or thousandths of minutes or hours because of the lack of units).
8) The Conclusion needs revision. Parts of it are highly redundant. It should not be a repetition of the Abstract.
9) Maximum feeding rate. Isn't it surprising that the asymptotes did not differ with such large differences in size? Is this likely to be a real pattern or an result of lack of power in your study? Obviously, you have to be careful about implying that a lack of difference means that they are the same, but it is a topic worth considering. Because looking at differences in maximum consumption was one of the topics of objective 2, this might be considered in its own paragraph as part of the discussion of objective 2.
10) I agree with Reviewer 1 that it would be highly desirable to show the data on which the functional response curves are based as well as the fitted curves. This would require Fig. 1 to have a panel for each size class and perhaps a fourth panel showing the confidence intervals to clarify the degree of overlap.

Other Comments

L37. Are you sure that the search coefficient can be called search efficiency? I think it may be a search rate.
L70. You need to define functional response (independent and dependent variables) at the start, to make sure that readers understand what you are referring to. As noted by Reviewer 2, a brief general definition of attack rate and handling time are also needed (the operational definitions for your specific study will be in methods).
L155. It would be useful to provide actual prey densities (number/volume). While a reader can calculate this if the 500-L mesocosms were full, this has not been specified and an actual density would be helpful for comparisons with other studies and the densities in the videorecorded aquarium trials.
L157. It would be useful to have more information on the mesocosms, such as their depth, light regime and temperature, whether water was circulated or still and whether any attempt was made to assess whether the mosquito larvae aggregated. I assume the larvae spent most of their time on the surface, which should probably be mentioned. My experience is that Xenopus mostly stay on the bottom. Do they therefore travel up to the surface to feed? Were the mesocosms located in such a way to avoid outside disturbances which might have affected mosquito surface use or Xenopus foraging?
L169ff. More information is needed here, including sample size, the order of introducing predator and prey to the aquarium, any acclimation of predator to aquarium, temperature and circulation of water, level of potential disturbance, distance of recording camera, camera speed. The information in L201-214 is video analysis (the entire paragraph except the statistical description at the end) and belongs in this section after it is corrected/clarified. If you did not acclimate the predators to the aquarium, you need to consider this as a possible explanation of some of your unexpected results.
L171-172. Indicate which dimension is which, depth in particular, for example providing L x W x D after the numbers. Also indicate the volume of water contained and provide actual densities (number/volume) for comparison to the mesocosm. Should you consider the implications of the much higher density for your observations than for your functional response estimation in your Discussion?
L195. Provide the units for the variables in the equations.
L234. For handling time, present as a sentence that indicates which sizes were different and which were not, as well as the trend in the data because you discuss this quite a bit.
L250-265. The first paragraph of the Discussion requires substantial revision. The first part (through L254) is highly redundant of the Introduction and does not need to be repeated. If you still feel that the assumption of homogenous predators needs to be discussed after reading Reviewer 1's comments, it should build on the Introduction with additional information and insights and come late in the Discussion. The next part of this paragraph summarizes conclusions that have not yet been established and therefore require additional discussion. For example, the U-shaped function of handling time was not supported statistically in the Results, so you need to explain why the trend may be real despite the lack of significance between medium and large frogs. Similarly, describing the relationship between search efficiency and size as inversely proportional ignores the lack of difference between medium and small and introduces search efficiency as a term previously considered as search coefficient and not justified as efficiency. I suggest introducing the relevant results briefly in the paragraph that discusses them and providing a more general statement here such as that both modeling prey consumption rates in relation to density and direct observation indicate that components of predation do change with size, at least between the smallest and the other two size classes.
L327. Indicate the species of predator and prey studied.
L339. Shouldn't feeding rate be higher if handling time is lower?

I have provided an annotated pdf with numerous suggestions for minor corrections to improve grammar, clarity and conciseness.

·

Basic reporting

1.A. The article reports on the results of mesocosm experiments to determine the functional response on Xenopus laevis frogs on one size class of larvae of a mosquito species. The basic presentation is good, although there are a number of small points where wording could be improved or there is lack of clarity; these are listed below. The main substantive point here is that it is not clearly stated that a single predator individual is used (?) in each mesocosm (this should be in the first paragraph of experimental procedure). If this is the case, there should be some quantification of between-individual variation in numbers eaten at each of the five prey densities for at least one (and maybe all three) predator size classes.
1.B. Specific corrections
line 70 Holling not Hollings (and maybe also note that he proposed a type-4, which has in fact been observed in many studies (Jeschke review), and predicted in a variety of scenarios (Abrams 1989).
line 79 – Note that in addition, a physical refuge in limited supply can create a type III response.
line 80-82 This sentence is too long and disjointed-rewording needed
line 155 Maybe add some justification for range of prey densities and for the distinct and narrow size class ranges.
line 259 "…data show a treat similar to…" would be better
line 264- 'is' should be 'in'
line 304 Don't you mean 'predator' rather than 'prey'
line 330 'which' should be 'with'??
line 332 'their' should be 'the'
line 335 Delete 'a' before 'higher'
line 339 –Doesn't a lower handling time mean a HIGHER feeding rate?

Experimental design

The design of the experiments is fairly traditional for functional response studies. However, this design can be misleading, and, as noted above, the presentation did not make it totally clear that there was one predator individual per mesocosm (I am assuming this, as no numbers were given). There was also no data presented on proportion of prey eaten. This is important for estimating parameters (using the equation on line 193) to be used in an expression (the functional response) that assumes a fixed density. The 48 hour starvation period can have different effects on different size classes; certainly it is likely that a large individual would have more energy reserves and would be less motivated to forage at a high rate. It would have been informative to examine responses over periods longer than the 14 hours used here.
Theory predicts that variation in foraging time or effort can be a large component of the effective 'handling time' in a type II response (Abrams 1991), and certainly digestion is likely to play a role. Actual physical handling time has been shown to underestimate 'h' relative to the 'h' obtained by fitting a response curve in several studies. There should have been some comparison of the parameter estimates based on actual consumption and those based on the video analysis.
It is not clear how an 'encounter' was identified.

Validity of the findings

The manuscript is structured around the theme that using a single predator size class to determine the functional response may be misleading in terms of the predator's effect on the prey. I do not think this is a good message. In the first place, it is widely known that size affects functional response parameters. The claim that "Often predators of the same species are assumed to be functionally equivalent" (line 28, 43 and later—e.g. line 251) is not backed up with references. People who work populations having significant size variation are unlikely to assume all sizes have the same functional response. It is true that classical models assume this (line 57), but they also assume that every predator is identical in EVERY respect, and it is not clear that the assumption of identical functional responses is worse that the assumption of identical numerical responses for all individuals. The reality is that these simple models are not meant to make quantitative predictions about any particular system. Theoreticians who have worked on size-structured models of predator and prey (e.g. deRoos and Persson book) have not assumed identical functional responses in all size classes. Using this argument is particularly problematic here in that the prey also has size structure, but only one size class is used in the experiments. If one wanted to build a predictive model, one would want to know susceptibility to predation of all prey sizes as well. This issue was examined in detail by McCoy et al (2011), an article that is mentioned briefly here and shares an author with this manuscript. It would be interesting to have some discussion of the combined implications of size structure in both species or speculation about which is more important for modeling.
3B. The two main findings are the type 3 response in the two larger size classes, but not the smallest, and the change in attack and handling parameters with size. There could be a more general discussion of hypotheses that could account for the type-3 response. The frogs are likely to be balancing costs and benefits of foraging, which is a major mechanism that could produce type-3 responses.
3.C. The effective functional response of a population having individuals with varying parameters is not the same as a response based on all individuals having mean parameter values unless it is linear (Chesson). It would therefore be of interest to see the range of individual capture rates at each of the densities (a plot with actual consumption rates of all individual's rates given as dots above each of the 5 prey density treatments in fig. 1)
3.D. Various other small issues
line 270 – It does not seem that handling time necessarily increases with larger predator size.
line 283 – Low enough densities of any prey will fail to provide enough reward for foraging. (Sarnelle et al 2015 CJFAS is an example—several theoretical works predicted this much earlier) This lower threshold should logically be lower for smaller individuals.
line 237 I did not see any analysis of how parameters based on video analysis differed from those based on data fitting; shouldn't this be present?
line 297 – The claim here seems to assume synchronous development of all frogs; is this the case? If not, the level of synchrony may be needed to estimate the effect on prey.
l. 299- A more accurate sentence would say 'There are some examples of a unimodal relationship between attack rate and predator size.'
l. 311 – It seems likely that attack behaviors/rates are adapted to the spectrum of prey sizes normally encountered (a broader range and no doubt different mean than in your experiments).
l. 335 and rest of paragraph – Whether a given feature is stabilizing in nature depends on the full response, which includes many other species and size classes—I'd recommend dropping the discussion of stability.
line 355 – A claim about many studies doing something should be backed up by more than one reference or it should be clear that the study referred to is or contains a review of many studies. I have not read the two Dick et al. studies referred to in the paper, so a wording change may be enough. However, standard ecological theory implies that the resource requirements and degree of generalization of the predator are much more important than the shape (or even mean value across prey densities) of the functional response in determining the impact of an invasive species.

Additional comments

The authors are right in claiming that few studies examine the predator-size dependence of its functional response, and it would be good to have this example published. However, the conclusions reached are often stated too broadly, and they should be qualified by noting the absence of other prey types and sizes, lack of examination of predator-dependence and other-species dependence of the functional response (see following paragraph). There also needs to be some discussion of the possibility that the pre-experiment 48 hour fast affected size classes differently and the possibility that the lowest prey density was not sufficiently low for detection of a type-3 response in the smallest size class.

This article had more references to old literature than most I've reviewed recently—this is good. However, there were some important ones left out. For example, there are older studies on size dependence of responses than the ones listed on line 64; the Eveleigh and Chant study in the references is one of these. The traditional approach to functional responses approaches this relationship as a rate of feeding by an isolated predator on a single prey population. This has been widely criticized, and there is general acceptance that predator density frequently influences the response (DeAngelis et al 1975; Abrams and Ginzburg 2000; Abrams 2015). In addition, the presence of higher level predators, other prey, animals that may be confused as prey, etc. all affect predator rates of feeding, and are therefore parts of any functional response that could be used in a predictive model of population dynamics. There should be some acknowledgement that what is dealt with here could easily be very different from what the relationship would be in the presence of other prey, higher level predators, etc.

Reviewer 2 ·

Basic reporting

See PDF for notes on clarity.

Experimental design

no comment

Validity of the findings

See PDF for notes on the discussion and conclusion.

Additional comments

Manuscript number: 24758
Full title: Size-dependent functional response of Xenopus laevis on mosquito larvae
Article type: Primary research paper

Reviewer overview:
This manuscript focuses on the dependence of functional responses on predator
body size. Specifically, the authors measure feeding rates of three size classes of
African clawed frogs on different densities of mosquito prey. They found that
functional response type was dependent on predator size. Additionally, both
observational data and parameter estimates found body size dependence of attack
rate and handling time. This work contributes to our understanding of intraspecific
variation in predator feeding rates and its potential impact on prey populations.
Overall, the research is interesting and well designed.

Major concerns: It is not clear why the authors use both observations and model
estimates to determine attack rate and handling time. The manuscript may benefit
by expanding the introduction to include predictions of the observational data and
rearranging the discussion to make clear links between parameter estimates
derived from models and mechanisms from observations of predator behavior.

Minor concerns:
Line 61, 94: effect of prey size on predator-prey interactions is outside the scope of
this manuscript and does not need to be included in the introduction.
Line 63: “these [studies] highlight”.
Line 72: define attack rate and handling time.
Line 74: Type II functional response is destabilizing as predators are unable to
regulate prey populations at densities beyond predator satiation.
Line 82-85: To someone unfamiliar with functional response models, it is unclear
that parameter estimates derived from models fit with data on feeding rates is not
the same as estimating parameters from observational data on behavior i.e. how are
the measurements of attack rate and handling time different between the two
methods? Do they provide different expectations? How does observational data
provide mechanistic understanding of variation in attack rate and handling time?
Line 88: Environmental conditions are outside the scope of this manuscript and do
not need to be included in the introduction.
Line 92: Prey size is not tested in this experiment and should not be in the
introduction as it leads the reader to believe that prey size will be part of the
experimental design.
Line 98: hypotheses for the effect of predator body size on handling times are
presented but there are no hypotheses for the effect of predator body size on attack
rates, unless “smaller predators…limited by” means predator attack rates are
constrained by predator body size.
Line 102: broader range of prey species or prey sizes?
Line 132-134: repetitive – edit for clarity. E.g. “temporary surface-water habitats”
and “colonize newly formed aquatic habitats”.
Line 149: “to quantify the functional response…dependent on consumer body size”.
Line 152-153: are these size classes the same life stage? If not, can you differentiate
the effect of body size vs life stage (Rudolf and Rassmussen 2013)?
Line 195: is there a difference between attack rate and the search coefficient? Table
2 and Figure 2 show search coefficient as used in equation 1, however the research
question posed in line 114 focuses on attack rate.
Line 240: What about the other components of handling time?
Line 244: What is scooping behavior?
Line 260: the phrase “handling efficiency” is confusing since “attack efficiency” is a
variable measured in the experiment and “handling efficiency” is a description of
predator behavior.
Line 287: contradicts line 244. All predators exhibited scooping behavior, but the
smallest predators exhibited sweeping behavior.
Line 288: contradicts findings in table 3.
Line 292-295: remove.
Line 297: Does level of predator ontogeny refer to body size or life stage?
Line 307: Additional size classes on the lower end?
Line 312-318: remove. Move lines 318-324 under earlier paragraph on predator
behavior.
Line 328-329: confusing- “smaller predators being more limited by handling times
than large prey”.
Line 342: high densities of predators can lead to increased intraspecific competition
and subsequent changes to an individual’s functional response, which can also be
size-depedent.
Line 343: Unclear how X. gilli is relevant to this study.
Line 359-362: the conclusion is not the place for new information. Move to
discussion.

Annotated reviews are not available for download in order to protect the identity of reviewers who chose to remain anonymous.

---

## Round 0.2 · Minor Revisions

Thank you for undertaking a careful and thorough revision. Both reviewers consider that your revised version is substantially improved and that it is requires only a few minor changes for improved clarity or accuracy of your statements. I agree with this evaluation, and have a small number of additional suggestions of my own.

Editor’s Comments
L1. Would adding ‘feeding on’ to the title make it clearer? “Size-dependent functional response of Xenopus laevis feeding on mosquito larvae”
L162. You apparently planned to insert a reference (which is needed) but forgot to do so.
L256. Pursuit is the foraging component that occurs after a prey is encountered and detected and the decision is made to capture it. It is search speed that affects the encounter rate.
L487. Repeated word.
Table 1, L3 p-values (hyphen)
Table 2. Provide units for b and h. This is important to allow other researchers to compare their results with yours.
Fig. 1. In the caption and x-axes provide the units for density. In the caption specify the length of the time period over which the number of prey captures occurred.
Fig. 2. Provide units for search coefficient and handling time in the caption and on the y-axes.

·

Basic reporting

Basic reporting is good. A few ambiguities are point out in the comments.

Experimental design

The experimental design is appropriate.

Validity of the findings

The findings are well justified.

Additional comments

line 72-73. This phrase; 'proportion of prey consumed is not constant', is vague and many interpretations consistent with this wording would be inaccurate. Would be better to say; 'the rate of prey consumption by a predator divided by prey abundance declines with prey abundance', which is the defining characteristic of the type 2 response. Although handling time is a common reason for this, it is not the only one (see Abrams 1990 Ecology). It would be better to say h is a measure of the nonlinearity of the response, and it corresponds to the amount of time required to handle a capture prey when an inability to capture prey while handling is the cause of the nonlinearity.

line 74 – 'due to high consumption rates at low densities' is not correct. The destabilization is due to the positive feedback on prey population growth caused by decreased predator consumption rates as a prey population increases.

76-77 Type three is defined by an accelerating increase in prey capture with increasing prey density for a range of low prey densities. (all responses have low capture at low density)

84-85 I'm unclear about what this last sentence contributes.

214-215 This is confusing- is there a difference between 'large' frogs and 'largest size class'? I think you mean 'medium' rather than 'large'

227 –'often assumed to be equivalent' This would be stronger with some examples. It is obviously true in simple models, but these assume everything is identical between individuals.

247 "BOTH zooplankton and…"

265 There is also a possibility that the relative fitness gain from small prey items is too small to make it worthwhile for larger foragers to be active.

281 The paragraph beginning here is somewhat long and you might want to split it with potential explanations for why the observed difference in F.R. shape occurred in the second one. In that respect Abrams 1991 Ecology and Abrams and Rowe 1996 Evolution both suggest that adults should take fewer risks and are more likely to have accelerating responses at low prey densities—basically because survival is more important relative to growth in adults than in young, which must grow quickly to escape a life stage with high unavoidable mortality. This was the point I was making with the line 283 comment (Sarnelle reference) in my earlier review.

Since there is a good deal of text about the importance of functional responses for applied questions, it might be good for the authors to acknowledge that shape of a response with respect to one prey is known to be affected (in theory, empirically or both) by the presence and abundance of other prey and of predators. Even Holling's and Murdoch's early work discussed type 3 arising from the presence of alternative prey. The response letter indicates the authors think that this is too big a topic to deal with; I think the acknowledgement of the limitation of results to the response to a single prey is important in this case of a generalist predator.

line 329-I think I signed the original review-Peter Abrams

Reviewer 2 ·

Basic reporting

The authors focus on quantifying the size-dependence of the functional response, a growing interest in food web ecology and important for understanding intra-specific functional diversity. The authors present sufficient background that are generally appropriate for the context of this paper. The following lines can be improved for clarity (line number based on word document with track changes by the authors):

ln 68: Still missing definitions of functional response, attack rate, handling time, therefore found this sentence out of place.

ln 68: This sentence may be better suited in the following paragraph (ln 76) since the paragraph following the next one (ln 103) reviews some of the causes of variation in functional forms. This is probably the most important sentence of the intro as it highlights whats missing in functional response lit and how this study seeks to fill that gap.

ln 76: Found this sentence in the rebuttal document - “The functional response is the key relationship linking predator and prey dynamics and describes the relationship between a predators uptake of prey as a function of the prey density.” Did not see this in the PDF or Word document with track changes. This sentence defines the functional responses whereas a previous version of the sentence did not.

ln 110: I’m not sure the connection between this example and your experiment, or with the previous statement if the focus of this experiment is varying predator size while keeping prey size constant. Is it that a predator can have the same prey over its lifetime, but there’s an optimal predator:prey size that maximizes feeding rates, and therefore preferences on prey size (Brose 2010, but see Uiterwaal et al 2017)? Maybe this point is better suited in the discussion.

ln 172: Not sure if this is correct wording. You’re estimating the form of the functional response; you’re using the functional response to quantify the effect of predator body size on prey consumption

ln 189: remove during?

ln 282-288: the authors state that this reference shows the form of the functional response changes with prey size, but proceed to describe predator size. Is this a typo, or did the reference alter both predator and prey size?

ln 489: Do smaller predators have higher handling times than larger predators or do they have longer handling times on large prey?

ln 502: This sentence is confusing following the previous sentence following prey densities. Also, are you trying to say that having a lot of predators means that they’re going to control prey populations i.e. what about predator dependence of functional response?

ln 520: Invasive predators aren’t mentioned in the intro or in the methods, so this statement was confusing to me.

Experimental design

With the removal of the video component, the aims, methods, and results are more clear in this version. A few comments to clarify the experimental design:

ln 115: predictions on the relationship between body size and handling time are presented, but missing prediction on the relationship between body size and attack rate.

ln 165: Unclear the range of volume used in the experiment. ± what volume?

ln 225: What indicates a type I response?

ln 238: The intro refers to attack rates, the methods refer to search coefficient. How are these related?

Validity of the findings

A few comments to clarify the authors' findings and add to the discussion:

ln 249: more appropriate for methods section, not results

ln 302: change "different from" to "different between"

Figure 1: Discuss possible reasons for the large variation in feeding for large predators at the highest prey density.

ln 363: authors state that increased handling times may be due to prey being too large to ingest via suction, but later state that this is not the case in their study (ln 467). As this paragraph is a discussion about handling time, moving this observation and including a discussion on its implications on the experimental results to this paragraph would be more appropriate.

ln 463: can you discuss possible reasons for the similarity in the search coefficient between medium and large predators?

ln 475: Might be more appropriate to discuss after ln 453 as it mentions predator detection.

ln 477: Does detection ability/success vary between the different size predators?

Additional comments

In the conclusion, the authors state that it is important to measure the functional response using different prey. An interesting question would be how prey traits affect the functional response i.e. is it prey size alone that affects the functional response, meaning different species of the same size are functionally similar, and therefore predator-prey relationships are predictable based on this trait, are other traits important, or do other traits interact with body size? A focus on traits, rather than species itself, may lead to generalities in predator-prey interactions.

Overall, well written, clear aims, and interesting findings.

---

## Round 0.3 · accepted · Accept

I apologize for the delay in making a decision on your manuscript. It arrived with several others while I was on vacation.

The manuscript is now ready for publication. I have attached a pdf with highlights indicating a few minor corrections.

#